# Effect of Clinical and Laboratory Parameters on HDL Particle Composition

**DOI:** 10.3390/ijms24031995

**Published:** 2023-01-19

**Authors:** Christina E. Kostara, Eleni T. Bairaktari, Vasilis Tsimihodimos

**Affiliations:** 1Laboratory of Clinical Chemistry, Faculty of Medicine, University of Ioannina, 45110 Ioannina, Greece; 2Department of Internal Medicine, Faculty of Medicine, University of Ioannina, 45110 Ioannina, Greece

**Keywords:** lipidomics, HDL, healthy individuals, aging, gender, menopausal status

## Abstract

The functional status of High-Density Lipoprotein (HDLs) is not dependent on the cholesterol content but is closely related to structural and compositional characteristics. We reported the analysis of HDL lipidome in the healthy population and the influence of serum lipids, age, gender and menopausal status on its composition. Our sample comprised 90 healthy subjects aged between 30 and 77 years. HDL lipidome was investigated by Nuclear Magnetic Resonance (NMR) spectroscopy. Among serum lipids, triglycerides, apoAI, apoB and the ratio HDL-C/apoAI had a significant influence on HDL lipid composition. Aging was associated with significant aberrations, including an increase in triglyceride content, lysophosphatidylcholine, free cholesterol, and a decrease in esterified cholesterol, phospholipids, and sphingomyelin that may contribute to increased cardiovascular risk. Aging was also associated with an atherogenic fatty acid pattern. Changes occurring in the HDL lipidome between the two genders were more pronounced in the decade from 30 to 39 years of age and over 60 years. The postmenopausal group displayed significant pro-atherogenic changes in HDLs compared to the premenopausal group. The influence of serum lipids and intrinsic factors on HDL lipidome could improve our understanding of the remodeling capacity of HDLs directly related to its functionality and antiatherogenic properties, and also in appropriate clinical research study protocol design. These data demonstrate that NMR analysis can easily follow the subtle alterations of lipoprotein composition due to serum lipid parameters.

## 1. Introduction

High-density lipoproteins (HDL) encompass dynamic multimolecular and multifunctional lipoproteins with biological roles that extend beyond their most recognized anti-atherogenic function, known as reverse cholesterol transport (RCT) [1]. Data supports that the functional status of HDLs is not dependent on the cholesterol content of the particle per se, measured as HDL-C, but rather it is inextricably related to their quality as determined by the morphological characteristics (shape, size), particles’ structure and stability, as well as by the overall lipid and/or protein composition [2]. Aberrations in the HDLs’ lipid cargo influence their stability, and, therefore, the eventual functional capability of these particles.

Cardiovascular disease, metabolic abnormalities and acute or chronic inflammatory milieu contribute toward the vulnerability of HDLs to compositional and structural modifications, causing disproportionation into two or more of their components [3,4,5,6]. As a result, the anti-atherogenic HDL particles can be converted to dysfunctional, and even pro-atherogenic and pro-inflammatory noxious equivalents [7], a phenotype, which may increase the cardiovascular risk.

However, before investigating the alterations of HDL lipid composition occurring in pathological conditions, it is important to (a) define the “normal” HDL lipidome and (b) understand how various intrinsic (e.g., age, gender, hormonal status) and extrinsic (e.g., diet, smoking, stress, exercise, biologic fluid used in the assays, i.e., serum or plasma, etc.) biological factors can trigger changes in HDL lipidome, and, thereby, in the HDL-related functionality in healthy individuals. Little information is available on the impact of the above-mentioned factors on HDL lipid composition and function in healthy individuals. This knowledge is not only of interest to better understand the HDLs’ remodeling capacity it but also could help in appropriate clinical research study design, e.g., for the definition of participants’ selection criteria.

Lipidomic analysis of HDLs has become an indispensable approach to gain in-depth characterization and interpretation of the lipid modifications occurring under physiological processes or pathological conditions, which may render them dysfunctional [8,9,10]. Knowledge of the mechanisms involved in HDL loss of function will be determinant to restoring the cardiovascular protection afforded by HDLs. Advances in analytical techniques such as nuclear magnetic resonance (NMR) spectroscopy and mass spectrometry (MS) have facilitated efforts to unravel disturbances in lipid molecules carried by HDLs beyond traditional cholesterol content measurements [11,12,13]. NMR-based lipid analysis, despite its lower sensitivity compared to MS and being typically limited by overlapping signals, is a non-destructive tool with high analytical reproducibility that does not require extensive steps for sample preparation, and offers a rapid, simple and non-invasive detection method of molecular lipid moieties and direct quantitative information [14].

In the present study, we used this lipidomic analysis for the detailed characterization of lipids extracted from HDL lipoprotein particles in the healthy population by ^1^H NMR spectroscopy. We investigated the association of the serum lipid profile with HDL lipid components, and the effect of the intrinsic factors aging, gender and menopausal status on the HDL lipidome.

## 2. Results

Characteristics of the study’s healthy participants: The total serum lipid parameters of the study healthy population are outlined in Table 1. Total cholesterol ranges from 117 to 200 mg/dL, triglycerides from 40 to 150 mg/dL, HDL-C from 25 to 71 mg/dL, LDL-C from 54 to 160 mg/dL and non-HDL-C from 63 to 171 mg/dL. The concentrations of apoAI and apoB are also reported, as well as the ratios HDL-C/apoAI, LDL-C/apoB and non-HDL-C/apoB.

HDL lipidome of healthy participants: Table 2 displays the percentage and the range of the major surface and core lipid molecules of HDLs of healthy participants. Cholesterol (total (TC), free (FC), and esterified (CE)), phospholipids (PLs), triglycerides (TG), total core and surface lipids, as well as the ratios CE/TC, CE/FC, CE/TG and TC/PLs were determined by the single NMR experiment. As expected, CE was the predominant form of cholesterol in HDLs, and the proportion of CE/FC was estimated to be about four (Table 2). PLs that quantitatively predominate in the HDL lipidome (54 ± 1.8, expressed in percentage of total HDL lipids) were subclassified in glycerophospholipids (GPLs), ether glycerolipids (ether GLs), and sphingolipids (SLs). Individual PLs molecules such as phosphatidylcholine (PC), lysophosphatidylcholine (lysoPC), phosphatidylethanolamine (PE), phosphatidylinositol (PI) and sphingomyelin (SM) have been quantified from characteristic well-resolved signals in the proton NMR fingerprint. PC, the major GPLs of HDLs, accounted for about 33% of the total lipids, whereas SM, the major SL class, was estimated at about 6.5% (Table 2). The rest GPLs are mainly attributed to phosphatidylserine (PS) and cardiolipin, the rest ether GLs to the platelet-activating factor (PAF), and finally the rest SLs are mainly attributed to ceramide, sphingosine-1-phosphate (S1P). The molecular ratio of PC/SM was calculated to be nearly five.

The fatty acid compositional pattern of esterified HDL lipids is shown in Table 3. The quantification of saturated (SFA) and unsaturated fatty acids (UFA) (total, monounsaturated (MUFA) and polyunsaturated fatty acids (PUFA)) was feasible, as well as the ratios SFA/UFA and SFA/PUFA. In addition, the individual PUFA, such as linoleic acid (LA) and the sum of Δ^5^ fatty acids, i.e., eicosapentaenoic + arachidonic acid (EPA and AA) and of docosahexaenoic acid (DHA) were quantified by their distinct signal. The percentage of LA esterified FC molecule was about 19%, followed by the sum of EPA + AA which was estimated at about 11%. Finally, the average chain length (ACL) of fatty acids was also calculated.

Impact of serum lipid parameters on the HDL lipidome in the whole healthy population: Table 4 shows the correlations between serum lipid parameters and HDL lipidome. Total serum cholesterol did not show a remarkable influence on HDL lipid composition, apart from a marginally significant positive correlation with the rest GPLs and the ratio of PC/SM (r = 0.23, *p* < 0.05 and r = 0.22, *p* < 0.05, respectively) and a negative correlation with plasmalogens PLs (r = −0.21, *p* < 0.05). Serum triglycerides levels had a stronger influence on HDL lipidome compared to that of total cholesterol levels; they were positively correlated with the percentage of TG (r = 0.54, *p* < 0.001), plasmalogens (r = 0.26, *p* < 0.01) and the rest SLs (r = 0.31, *p* = 0.003), and negatively correlated with CE (r = −0.23, *p* = 0.03), PE (r = −0.26, *p* = 0.02) and SM (r = −0.39, *p* < 0.001). These changes subsequently influenced the ratios CE/TC (r = −0.20, r = 0.04), CE/FC (r = −0.23, *p* = 0.03, CE/TG (r = −0.62, *p* < 0.001) (Table 4). Serum HDL-C did not show any remarkable influence on HDL composition apart from a marginally significant negative correlation with PE (r = −0.22, *p* < 0.05) and a positive association with sphingolipids (r = 0.28, *p* < 0.01) and the rest SLs (r = 0.31, *p* < 0.01). The concentration of apoAI and the ratio HDL-C/apoAI had a stronger effect on HDL lipid composition compared to that observed for HDL-C. ApoAI levels were positively associated with TG (r = 0.34, *p* = 0.003), rest SLs (r = 0.33, *p* = 0.004) and the ratio PC/SM (r = 0.28, *p* < 0.05), and were negatively associated with PE (r = −0.31, *p* = 0.006), PI (r = −0.25, *p* < 0.05) and the ratio CE/TG (r = −0.33, *p* = 0.004). The ratio HDL-C/apoAI was positively associated with total PLs (r = 0.33, *p* = 0.004), total SLs (r = 0.26, *p* < 0.05), SM (r = 0.26, *p* < 0.05), surface lipids (r = 0.24, *p* < 0.05), and the ratios CE/TC (r = 0.32, *p* < 0.01), CE/FC (r = 0.33, *p* < 0.01) and CE/TG (r = 0.41, *p* < 0.001), and were negatively associated with FC (r = −0.31, *p* < 0.01), TG (r = −0.43, *p* < 0.001), lysoPC (r = −0.28, *p* < 0.05), plasmalogens (r = −0.23, *p* < 0.05), core lipids (r = −0.24, *p* < 0.05) and the ratio TC/PLs (r = −0.26, *p* < 0.05) (Table 4).

Concerning the atherogenic lipoproteins, LDL-C did not show any remarkable influence on HDL lipidome apart from a positive association with the rest GPLs (r = 0.25, *p* < 0.05) and the ratio CE/TG (r = 0.33, *p* = 0.001) and a negative association with TG (r = −0.31, *p* = 0.003) and plasmalogens (r = −0.26, *p* = 0.01). The only effect that non-HDL-C had on HDL lipidome was a marginally positive correlation with the rest GPLs (r = 0.25, *p* < 0.05). ApoB slightly influenced the content in TG (r = 0.24, *p* < 0.05), lysoPC (r = −0.36, *p* < 0.001) and the rest GPLs (r = 0.32, *p* < 0.01).

The correlations between total serum lipid parameters and the fatty acid composition of HDL lipids are shown in Appendix A. Serum lipid parameters had no significant effect on HDL fatty acids apart from a slight effect of HDL-C/apoAI on DHA content (r = 0.27, *p* < 0.05).

Effect of aging on the HDL lipidome: We investigated the effect of aging on HDL lipidome of the total healthy population, as well as of both genders separately (Figure 1). Aging was associated with both qualitative and quantitative aberrations in the HDL lipidome that may contribute to increased cardiovascular risk (Figure 1). Specifically, the progressive increase in age was positively associated with TG (r = 0.62, *p* = 0.001), lysoPC (r = 0.39, *p* = 0.001), rest SLs (r = 0.52, *p* = 0.001) and FC (r = 0.52, *p* = 0.001). These changes subsequently influenced the ratio TC/PLs. On the other hand, the percentages of CE (r = −0.37, *p* = 0.001), total PLs (r = −0.36, *p* = 0.001), total GPLs (r = −0.30, *p* = 0.004) and those of individual PLs such as PC (r = −0.37, *p* = 0.001), PE (r = −0.72, *p* = 0.001), PI (r = −0.22, *p* = 0.04) and SM (r = −0.45, *p* = 0.001) decreased significantly with aging. As a result, these changes influenced the ratios CE/TC, CE/FC and CE/TG. The percentages of TC, rest GPLs, total ether GLs, plasmalogens phospholipids, rest GLs and the ratio of PC/SM remained stable with aging.

Aging was associated with an atherogenic fatty acid pattern of HDL lipids including enrichment in SFA (r = 0.34, *p* = 0.001) and depletion in UFA (r = −0.34, *p* = 0.001), mainly due to PUFA (r = −0.25, *p* = 0.02), LA (r = −0.49, *p* = 0.001), and DHA (r = −0.42, *p* = 0.001) (Figure 1). The ACL of fatty acids and the ratio of SFA/UFA were decreased with aging (r = −0.30, *p* = 0.005, and r = −0.31, *p* = 0.003, respectively), whereas the ratio of SFA/PUFA was increased (r = 0.28, *p* < 0.01). The percentages of MUFA and the sum of EPA+AA were not affected by aging.

It is worth noting that the majority of HDL lipids and fatty acids were altered to the same direction for both genders and most of the observed changes were statistically significant for women (Figure 1).

Gender effect on the HDL lipidome: In this analysis carried out to examine the presence of possible gender-dependent alterations in HDL lipidome, 44 men and 46 women had participated after adjustment for age and serum lipid concentrations (Appendix A). Our analysis did not reveal significant differences in the major HDL lipid classes, core and surface lipids, as well as the ratios CE/TC, CE/FC, CE/TG and TC/PLs (all *p* > 0.05) between the two genders (Table 5). However, we found statistically significant changes in individual GPLs between males and females including PC and PI that were significantly lower in females compared to males, while the percentage of the rest GPLs was significantly higher (Appendix A). No significant changes were observed in FA pattern between the two genders, apart from a significant increase in DHA in females compared to males (Appendix A). Then, we investigated whether the HDL lipidome was altered per age decade between the two genders. Although HDL-C levels did not present a statistically significant difference between the two genders aged from 30 to 39 years (45 ± 8 vs. 49 ± 9, *p* > 0.05), changes in specific lipids were observed. The percentage of FC was significantly lower in females compared to males, whereas total PL content was significantly higher (Table 5). The analysis of individual PLs showed that the percentage of PC was significantly lower in females compared to males, whereas that of the rest GPLs was higher (Appendix A). As expected, the aforementioned changes led to a significant increase in CE/TC and CE/FC, whereas TC/PLs was significantly decreased (Table 5). Concerning the FA pattern, both males and females presented similar and not significant changes, apart from a significant increase in MUFA in females compared to males (Appendix A).

The gender-related changes in the major lipid constituents of HDLs narrowed in the decades from 40 to 49 and from 50 to 59 years of age (Table 5). In subjects aged 40–49 years, among the various PLs measured in the NMR spectrum, only lysoPC and plasmalogens phospholipids were statistically different between the two genders (Appendix A). Specifically, the proinflammatory lysoPC was significantly higher in females compared to males, whereas plasmalogens phospholipids were lower (Appendix A). In the decade from 50 to 59 years of age, we observed disturbances in more PLs compared to those observed in the decade 40–49 years. Females tended to have a more pro-atherogenic HDL lipid profile compared to males. In particular, PC, PE and plasmalogens phospholipids were significantly lower in females compared to males, whereas the rest GPLs were significantly higher (Appendix A). Both males and females presented with similar FA pattern in HDL lipids (Appendix A).

Interestingly, over 60 years, the two groups presented remarkable changes in HDL lipidome notwithstanding the fact that HDL-C levels were not statistically significantly different. Females were presented with more atherogenic alterations in HDL lipidome compared to males. More specifically, they displayed a significantly higher percentage of core lipids compared to males mainly due to the significantly higher percentage of TG (Table 5). Additionally, HDLs’ surface was significantly depleted in lipids in females compared to males mainly due to the decrease in total PL content, while FC was increased. The decrease in total PLs was mainly due to the significant decrease in the GPLs, PC and PI (Appendix A). The percentage of proinflammatory molecule, lysoPC, was significantly increased in females compared to males. The above-described changes significantly influenced the ratios CE/TC, CE/FC and TC/PLs (Table 5) and the PC/SM ratio (Appendix A). Although total SFA and UFA content did not differ between the two groups, significant changes were observed in total content, as well as in individual PUFA such as LA, DHA and in the ratio SFA/PUFA (Appendix A).

Effect of menopausal status on the HDL lipidome: To investigate the effect of menopausal status on HDL lipidome, female subjects were classified into groups based on their menopausal history: 22 premenopausal and 24 postmenopausal. The biochemical characteristics of the two groups studied are outlined in Appendix A. As expected, the two groups were significantly different in age. Among the various parameters measured, TG, apoAI and the ratio non-HDL-C/apoB increased in postmenopausal females compared to premenopausal, whereas the ratio HDL-C/apoAI was decreased. It is worth noting that no significant change was observed in HDL-C levels between the two groups (Appendix A).

Postmenopausal females displayed a significantly higher percentage of core lipids in HDLs (38.65 ± 1.38 vs. 37.61 ± 1.12, *p* < 0.01) mainly due to the significantly higher percentage of TG (4.74 ± 0.89 vs. 3.40 ± 0.39, *p* < 0.001) compared to premenopausal (Figure 2). In postmenopausal females, HDLs appeared to be enriched in cholesterol (42.72 ± 2.07 vs. 41.38 ± 1.17, *p* < 0.05) mainly due to a significantly higher percentage of FC (8.81 ± 1.15 vs. 7.17 ± 0.61, *p* < 0.001), while CE was lower but without significance (33.90 ± 1.44 vs. 34.21 ± 0.95). The aforementioned changes subsequently resulted in a significant decrease in CE/TC, CE/FC and CE/TG ratios in postmenopausal females compared to premenopausal (0.79 ± 0.02 vs. 0.83 ± 0.01, *p* < 0.001, 3.90 ± 0.46 vs. 4.80 ± 0.40, *p* < 0.001 and 7.41 ± 1.48 vs. 10.20 ± 1.26, *p* < 0.001, respectively). Total surface lipid content was significantly lower in the postmenopausal group compared to the premenopausal group (61.35 ± 1.38 vs. 62.39 ± 1.12, *p* < 0.01) due to the significantly lower percentage of total PLs (52.54 ± 2.04 vs. 55.22 ± 1.32, *p* < 0.001) resulting in a significant increase in the TG/PL ratio (0.82 ± 0.07 vs. 0.75 ± 0.04, *p* < 0.001) (Figure 2). The significantly lower percentage of total GPLs (40.42 ± 2.04 vs. 42.78 ± 1.42, *p* < 0.001) is the main cause for the lower percentage of total PLs observed in postmenopausal females compared to premenopausal.

As shown in Figure 3, the profile of individual GPLs and SLs in postmenopausal females was different from that recorded for premenopausal females. The percentages of PC, SM, PE, and PI were significantly lower in the postmenopausal group compared to the premenopausal (31.21 ± 2.71 vs. 33.43 ± 1.09, *p* < 0.01, 6.18 ± 1.25 vs. 7.02 ± 0.66, *p* < 0.01, 0.77 ± 0.22 vs. 1.18 ± 0.11, *p* < 0.001 and 1.11 ± 0.33 vs. 1.42 ± 0.21, *p* < 0.001, respectively), whereas lysoPC and the rest SLs were significantly higher (3.65 ± 0.91 vs. 2.93 ± 0.31, *p* < 0.001 and 1.16 ± 0.78 vs. 0.33 ± 0.18, *p* < 0.001, respectively). No statistically significant differences were observed in the FA pattern of HDL lipids between the two groups, apart from a significant decrease in the percentages of LA and DHA in postmenopausal females compared to premenopausal (Appendix A).

## 3. Discussion

In the present study, we investigated the HDL lipidome in a healthy population by ^1^H NMR spectroscopy, and the influence of serum lipid profile, age, gender and menopausal status on its composition.

HDL lipidome comprises around half of the particle’s mass and lipids, the most abundant being phospholipids, sphingolipids, cholesterol, cholesteryl esters, and triglycerides [8]. Modifications in these structural components remodel HDLs affecting their functionality and antiatherogenic functions, which is an important determinant of the onset and progression of cardiovascular disease and its outcome [15]. Wide-ranging structural and compositional modifications have been reported in pathological conditions, many of which have been implicated in the generation of a HDL dysfunctional phenotype. To the best of our knowledge, a detailed analysis of the HDL lipid composition in healthy individuals, as well as the influence of intrinsic factors, such as age, gender and menopausal status on HDL lipid composition, has not been reported. A few studies have investigated the influence of these factors on individual lipid components of HDLs [16,17].

### 3.1. Impact of Serum Lipid Parameters on the HDL Lipidome

We observed that the magnitude of the relationship between the serum lipid profile and HDL lipid composition was greater for serum TG, apoAI, apoB levels and the ratio HDL-C/apoAI, while HDL-C levels did not show a remarkable influence on HDL composition.

Despite the fact that serum TG levels were within normal range, they were positively correlated with TG content in the HDLs’ core and were negatively correlated with CE and the ratio CE/TG. In addition, serum TG levels were negatively associated with the ratios CE/FC and CE/TC; the latter reflects the percentage of cholesterol esterification in the HDL fraction. Enrichment of the HDLs’ core in TG, together with simultaneous depletion in CE, constitutes a key mechanism involved in impaired HDL-cholesterol efflux capacity. The resulting HDLs are intrinsically more unstable and undergo rapid catabolism via the action of hepatic lipase [18]. In addition, these particles may have attenuated antioxidative activity because the aforementioned disturbance in the core considerably alters the conformation of the central and C-terminal domains of apoAI which are critical for HDL to act as an acceptor of LDL-derived oxidized lipids [18]. Since apoAI is the principle catalytic activator of LCAT, the disproportionation into CE and TG in the HDL core possibly negatively influence the LCAT-mediated conversion of FC to CE [19].

The fluidity of the surface monolayer is strongly determined by their amount and proportion in PL molecules and is particularly important for both scavenger receptor class B member 1 (SR-BI)- and ATP-binding cassette transporter A1 (ABCA1)-mediated cholesterol efflux [20]. The enrichment of HDLs with SM enhances the bidirectional flux of cholesterol from SR-BI expressing cells to HDL by decreasing the influx of cholesterol [21]. In addition, the ATP binding cassette transporter G1 (ABCG1) and the SR-BI-mediated efflux was higher when HDLs contain SM instead of PC [21]. Additionally, PLs are also responsible for the ability of HDLs to inhibit the cytokine-mediated increase in the endothelial cell expression of adhesion molecules [21], a process known as anti-inflammatory activity. Apart from PC, which was thought to be the main lipid component responsible for this activity of HDLs [22], HDLs containing SM were found to be more efficient in inhibiting the release of TNF-α, IL-6, and IL-1β compared to those containing PC [21]. Thus, the negative correlation of serum TG levels with SM content in the HDL surface could negatively affect the anti-inflammatory potential of HDLs.

On the contrary, in the state of an increased HDL-C/apoAI ratio, where the particle’s core is depleted in TG, the resulting increased CE/TG ratio has a beneficial effect on HDL stability and potentially results in a more effective apoAI function and a reverse cholesterol transport process. Moreover, the HDL-C/apoAI ratio was positively associated with PLs, CE/TC and CE/FC, and negatively associated with the pro-inflammatory molecule lysoPC. These data demonstrate that NMR analysis can easily follow the subtle alterations of lipoprotein composition due to serum lipid parameters.

### 3.2. Aging-Specific Changes in HDL Lipidome

It is well known that during the physiological process of aging, cardiovascular disease risk increases. Lipid metabolism has an active role in biological metabolic processes that contribute to healthy aging and age-related diseases [23], although the underlying mechanisms have not been fully elucidated [24]. Aging induces physicochemical modifications of the HDL structure that affect its functionality [25]; however, there are few published data on the influence of aging on HDL lipidome [17]. Our results showed that the HDL lipidome of healthy individuals was profoundly altered with aging toward an atherogenic lipid pattern that potentially renders HDLs dysfunctional. As age increases, HDLs were depleted in PLs and CE, and enriched in TG and lysoPC, changes that strongly affected the ratios CE/TC, CE/FC, CE/TG and TC/PLs. Moreover, a shift of fatty acids from an unsaturated to a saturated state was also observed.

As mentioned above, the enrichment of the HDLs’ core in TG together with the simultaneous depletion in CE results in intrinsically more unstable HDLs [26,27] with attenuated antioxidative activity [26,28] and impaired HDL-cholesterol efflux capacity. Berrougui et al. [29] found significant alterations in PL layer fluidity and apoAI conformation in elderly subjects when compared to young subjects, as well as a significant reduction in the PC/SM ratio which caused an impairment of HDL-mediated RCT capacity. The latter has been attributed to a reduction in the ATP-binding cassette transporter A1 (ABCA1) pathway [29]. Similar changes have been described by Holzer et al. [17]. Aging was negatively correlated with the ratios CE/FC and CE/TC; the latter reflects the percentage of cholesterol esterification in the HDL fraction. A positive association of aging with the proinflammatory lysoPC was also found in our study. Elevated lysoPC levels were positively associated with age-related pathologies, such as atherosclerosis, diabetes and ischemia [30].

It is well known that dietary intake and endogenous metabolic processes closely determine the fatty acid pattern of HDL lipids, and, thus, the fluidity of the surface monolayer, which, in turn, normally regulates the cholesterol efflux capacity of HDLs [31]. Apart from that, SFA have been linked causally to inflammation via the activation of Toll-like receptor (TLR)-mediated proinflammatory signaling pathways [32], increasing endothelial injury and impairing endothelial repair capacity [33], whereas PUFA improves endothelial function. Thus, the observed shift of fatty acids from unsaturated to saturated in our study could induce the activation of proinflammatory metabolic pathways during aging. The altered hepatic lipid metabolism, the expression of LDL receptors, and the intestinal uptake of lipids that are known to affect plasma lipid profiles are altered by aging [34,35]. Overall, the aforementioned age-related alterations occurring simultaneously in HDL lipidome could affect either the HDL particles’ stability or their functionality, by modulating surface charge, fluidity or binding to cellular receptors. These changes may be associated with increased vulnerability of HDLs to age-related diseases, and potentially explain a detrimental role of dysfunctional HDLs to the high burden of cardiovascular disease observed with aging.

### 3.3. Gender-Specific Changes in HDL Lipidome

The gender-related differences in HDLs have been investigated with women having a two-fold higher concentration of large HDL particles than men [36]. However, very little is known about the compositional differences in HDL lipidome between the two genders. Men were characterized by a less favorable lipoprotein–lipid profile, which was accompanied by a smaller HDL particle size compared to women [37]. According to Badeau et al., gender did not have a role in modifying the first step of reverse cholesterol transport as a protective mechanism against cardiovascular disease [38].

In our study, no significant gender-related differences were observed in HDL lipidome after adjusting for age, and serum lipid levels. Gender-specific differences in HDL lipidome were found in the decade of 30 to 39 years of age, and also over 60 years, whereas the gender discrepancy in the HDL lipidome narrows markedly in decades of 40–49 and 50–59 years of age. More specifically, the decade from 30 to 39 years of age for females exhibits a more favorable HDL lipid profile compared to the age-matched males, as illustrating from the lower levels of FC and the higher levels of PLs, CE/TC and CE/FC ratios compared to males. Of note, at over 60 years of age, we found that the alterations in HDL lipidome were more prominent and tended to be more pro-atherogenic in females compared to the age-matched males that thought to be, due, in part, to a deficiency of estrogen. Thus, in females, HDLs were enriched in TG and FC, and depleted in PLs, mainly due to a decrease in PC compared to males. These changes strongly affected the ratios CE/TC, CE/FC, TC/PLs and PC/SM.

### 3.4. Menopause Status-Specific Changes in HDL Lipidome

Finally, we investigated the alterations occurring in the HDL lipid composition in postmenopausal women compared to premenopausal women. To the best of our knowledge, there are few data in the literature on the lipoprotein profile of women in menopause transition, which mainly concerns the investigation of the plasma lipid profile or lipoprotein subclass profiles and sizes [39]. A shift toward reduced function per HDL particle was observed with large HDLs to become less efficient in promoting cholesterol efflux during the menopause transition [40].

In our study, the aberrations occurring in the HDL lipidome of postmenopausal group undoubtedly shifted to a more pro-atherogenic lipid profile compared to premenopausal. We found an enrichment of HDLs in TG, FC, and lysoPC, and a depletion in CE and PLs such as PC, PE, PI, SM, findings which could be attributed to the progressively attenuating function of endogenous estrogens [41]. It is well known that the risk of developing cardiovascular disease is markedly lower during the premenopausal than the postmenopausal period of life [42], primarily due to the presence of endogenous estrogens [43,44] which exert antiatherogenic effects [45,46]. Estrogen signaling pathways have pleotropic effects on many pathways that govern the lipid and lipoprotein metabolism, but our understanding of these effects is complicated [47].

## 4. Materials and Methods

Subjects: The group studied consisted of 90 healthy subjects (46 females and 44 males) aged between 33 and 74. The mean ± SD age was 49.5 ± 12.7. All subjects were non-smokers. No individual had evidence of cardiovascular disease according to history, clinical examination, or electrocardiograms. None of the participants were taking lipid-lowering drugs or any other medication known to affect lipid metabolism, including hormonal replacement therapy.

Sample Collection: Fasting venous blood samples were obtained in the morning after an overnight fast for all study participants. Serum was separated by centrifugation at 3000× *g* for 15 min for the determination of biochemical parameters, and one 1.5 mL aliquot was stored at −80 °C until NMR analysis.

Ethics statement: The collection of the samples from all participants was conducted in accordance with the guidelines of the Ethics Committee of the University Hospital of Ioannina. Written consent was obtained from each participant.

Determination of Biochemical Parameters: Serum lipid parameters were measured on an AU5400 Clinical Chemistry Analyzer (Beckman, Hamburg, Germany) by standard procedures. Total cholesterol and triglycerides were determined enzymatically and HDL-cholesterol was determined by a direct assay. LDL-cholesterol was calculated by the Friedewald formula (provided that triglycerides levels were lower than 400 mg/dL or 4.5 mmol/L) and non-HDL-cholesterol was calculated by the equation: non-HDL-cholesterol = total cholesterol-HDL-cholesterol. Serum apoAI and apoB were measured by immunonephelometry on a BN ProSpec System (Siemens, Marburg, Germany).

### 4.1. HDL Lipidome Analysis by NMR Spectroscopy

Isolation and lipid extraction of HDL lipoproteins: *HDL* particles were isolated from non-HDL particles by precipitation with dextran sulfate/MgCl_2_, and their lipid content was extracted according to the modified Bligh and Dyer method [48].

^1^H NMR spectroscopy: The extracted HDL lipids were dissolved in 500 μL of deuterated methanol/chloroform (2:1, *v*/*v*). All NMR spectra were recorded on a 500 MHz Bruker Avance DRX NMR spectrometer (NMR Center, University of Ioannina) operating at a field strength of 11.74 Tesla. A “zgpr” Bruker pulse program was applied with the parameters as follows: 64 scans, 64 K data points with a 5000 Hz spectral width, and a 90° pulse. All free induction decays (FIDs) were multiplied by an exponential weighting function corresponding to the 0.3 Hz line-broadening factor prior to Fourier transformation. NMR spectra were phase- and baseline-corrected and referenced to the methanol signal (δ = 3.30 ppm) using TopSpin 2.1 software (Bruker Biospin Ltd., GmbH, Rheinstetten, Germany). The quantification of HDL lipids was based on the integration of characteristic well-resolved signals in the NMR spectrum, corrected for the number of protons and then normalized with respect to the signal from the cholesterol C18 methyl group at 0.68 ppm. The lipid composition of HDLs were expressed as percentages of the total lipid content.

### 4.2. Statistical Analysis of Data

Data analysis was performed with SPSS software (version 23.0; IBM Corp., Armonk, NY, USA). All quantitative data are expressed as mean values ± standard deviation (SDs). Group comparison was performed using independent samples *t*-test and a *p* value of < 0.05 was considered to indicate statistical significance. Pearson correlation coefficients were used to describe the relationship of the serum lipid profile and age with HDL lipid composition.

## 5. Conclusions

Our results have shown that intrinsic factors, such as age, gender and menopausal status, remodel HDL lipid composition and should be carefully considered for the appropriate clinical research study protocol design. The mechanisms underlying these changes in HDL lipid composition, how these changes actually contribute and could explain the increase in the incidence of cardiovascular diseases observed during aging and menopausal time, and how reversing these changes might improve metabolic health remain relatively unknown. Further studies in larger independent cohorts are recommended to gain new insights into the mechanisms underlying the influence of intrinsic factors and healthy aging.

## Figures and Tables

**Figure 1 ijms-24-01995-f001:**
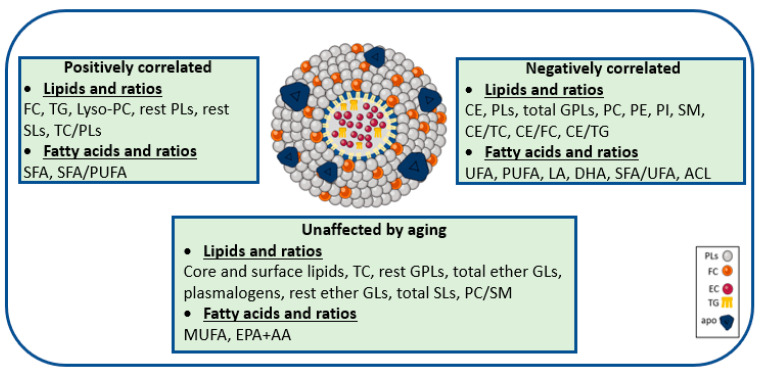
HDL lipidome and fatty acids positively or negatively correlated, or unaffected by aging. Key: ACL, average chain length; CE, cholesterol esters; DHA, docosahexaenoic acid; EPA + AA, the sum of eicosapentaenoic and arachidonic acid; FC, free cholesterol; GLs, glycerolipids; GPLs, glycerophospholipids; Lyso-PC, lysophosphatidylcholine; MUFA, monounsaturated fatty acids; TC, total cholesterol; TG, triglycerides; PC, phosphatidylcholine; PE, phosphatidylethanolamine; PI, phosphatidylinositol; PLs, phospholipids; PUFA, polyunsaturated fatty acids; SFA, saturated fatty acids; SLs, sphingolipids; SM, Sphingomyelin; UFA, unsaturated fatty acids.

**Figure 2 ijms-24-01995-f002:**
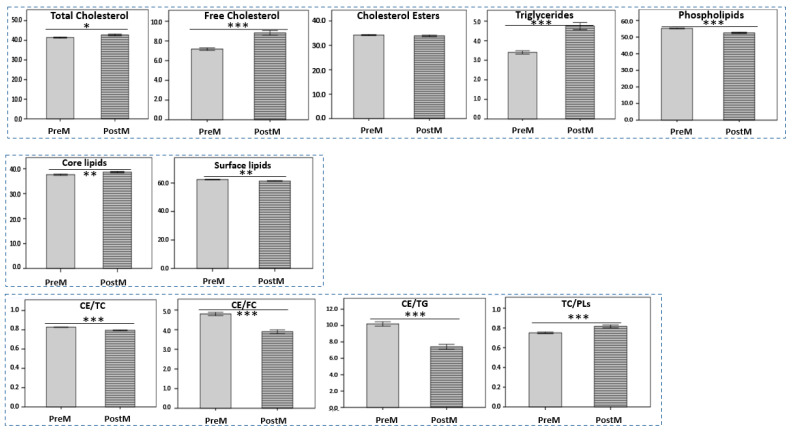
HDL lipid components and their ratios in the study groups (mean ± SD). PreM: premenopausal women; PostM: postmenopausal women. * *p* < 0.05; ** *p* < 0.01; *** *p* < 0.001: compared to the premenopausal group.

**Figure 3 ijms-24-01995-f003:**
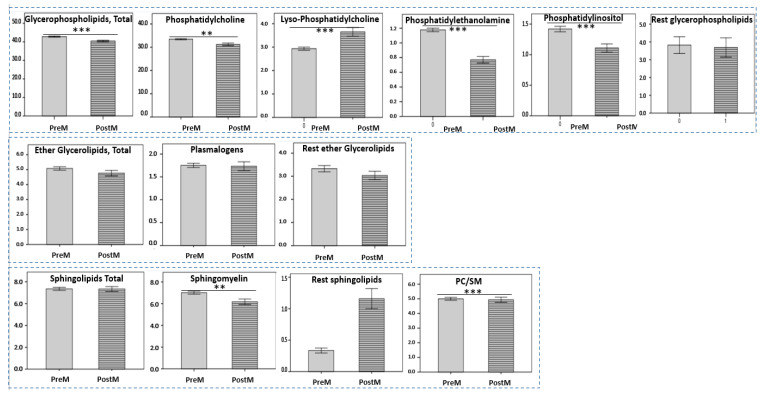
HDL phospholipid components and PC/SM ratio in the study groups (mean ± SD). PreM: premenopausal women, PostM: postmenopausal women. ** *p* < 0.01; *** *p* < 0.001: compared to the premenopausal group.

**Table 1 ijms-24-01995-t001:** Serum lipid parameters and apolipoproteins of the study population.

	Mean ± SD	Range
n	90	
Age, years	49.5 ± 12.7	30–77
Gender, M/F	44/46	
TC (mg/dL)	171 ± 20	117–200
TG (mg/dL)	96 ± 30	40–150
HDL-C (mg/dL)	49 ± 9	25–71
LDL-C (mg/dL)	101 ± 23	54–160
non-HDL-C (mg/dL)	122 ± 21	63–171
apoAI (mg/dL)	136 ± 25	80–188
apo B (mg/dL)	81 ± 17	39–116
HDL-C/apoAI	0.4 ± 0.05	0.3–0.5
LDL-C/apoB	1.2 ± 0.2	1.0–1.8
non-HDL-C/apoB	1.5 ± 0.2	1.2–2.2

**Table 2 ijms-24-01995-t002:** HDL composition in percentage of major lipid classes and phospholipid molecules in the whole healthy population.

Major Lipid Classes	Mean ± SD	Range
**Cholesterol, total (TC)**	**41.9** **± 1.6**	**38.1–45.8**
free (FC)	7.9 ± 1.0	6.5–11.4
esterified (CE)	34.0 ± 1.1	31.0–36.1
**Triglycerides (TG)**	**4.1** **± 0.9**	**2.3–6.5**
**Phospholipids (PLs), total**	**54.0** **± 1.8**	**49.3–59.5**
**Core Lipids, total**	38.0 ± 1.2	33.8–41.3
**Surface Lipids, total**	62.0 ± 1.2	58.7–66.2
**Ratio**		
**CE/TC**	0.8 ± 0.0	0.7–0.8
**CE/FC**	4.3 ± 0.5	3.0–5.2
**CE/TG**	8.7 ± 1.8	4.8–13.5
**TC/PLs**	0.8 ± 0.1	0.6–0.9
**Phospholipids**	**Mean** **± SD**	**Range**
**Total Glycerophospholipids (GPLs)**	**41.8** **± 1.9**	**36.8–46.2**
Phosphatidylcholine (PC)	33.1 ± 2.3	22.4–37.6
Lysophosphatidylcholine (LysoPC)	3.2 ± 0.7	1.8–5.2
Phosphatidylethanolamine (PE)	1.0 ± 0.2	0.4–1.5
Phosphatidylinositol (PI)	1.4 ± 0.4	0.4–3.8
Rest GPLs ^a^	3.1 ± 2.2	0.1–8.8
**Total Ether Glycerolipids (Ether GLs)**	**4.9** **± 0.7**	**3.1–6.7**
Plasmalogens	1.8 ± 0.3	1.2–3.1
Rest ether GLs ^b^	3.1 ± 0.8	1.1–5.3
**Total Sphingolipids (SLs)**	**7.3** **± 0.9**	**5.2–9.7**
Sphingomyelin (SM)	6.5 ± 1.1	2.9–8.5
Rest SLs ^c^	0.8 ± 0.4	0.1–4.8
**Ratio**		
PC/SM	4.9 ± 0.6	3.6–7.3

Values are expressed in percentages of total lipids (mol/100 mol of total lipid content) and are given as mean ± SD and range. Bold: Main lipid classes. ^a^: Mainly phosphatidylserine, phosphatidylglycerol; ^b^: Mainly PAF; ^c^: Mainly ceramide.

**Table 3 ijms-24-01995-t003:** Fatty acid profile of esterified HDL lipids in the total healthy population.

Fatty Acid Pattern	Mean ± SD	Range
** Saturated fatty acids (SFA) **	36.8 ± 6.6	20.0–51.6
**Unsaturated fatty acids (UFA)**	63.2 ± 6.6	48.4–80.0
**Monounsaturated fatty acids (MUFA)**	8.6 ± 4.9	0.1–22.1
**Polyunsaturated fatty acids (PUFA)**	54.6 ± 5.8	36.1–73.5
Linoleic acid (LA)	19.4 ± 2.8	13.4–27.3
Eicosapentaenoic + arachidonic acid (EPA + AA)	10.7 ± 1.6	6.1–16.0
Docosahexaenoic acid (DHA)	3.7 ± 0.7	2.3–6.0
**Ratio**		
SFA/UFA	0.6 ± 0.2	0.3–1.1
SFA/PUFA	0.7 ± 0.2	0.3–1.4
Average chain length (ACL)	16.2 ± 1.6	11.9–20.5

Values are expressed in percentages of total lipids (mol/100 mol of total fatty acids) and are given as mean ± SD. Bold: Different FA class.

**Table 4 ijms-24-01995-t004:** Correlation between serum lipid and apolipoprotein parameters and HDL lipidome.

HDL Lipidome	Serum Lipid and Apolipoprotein Profile							
	TC	TG	HDL-C	ApoAI	HDL-C/apoAI	LDL-C	non-HDL-C	Apo B
TC	0.07 *	−0.08	0.02	0.05	−0.16	−0.02	−0.06	−0.09
	-	-	-	-	-	-	-	-
FC	0.11	0.13	0.02	0.15	−0.31	−0.16	−0.11	−0.17
	-	-	-	-	0.006	-	-	-
CE	0.02	−0.23	0.05	−0.07	0.05	0.12	0.02	0.02
	-	0.03	-	-	-	-	-	-
TG	−0.08	0.54	0.08	0.34	−0.43	−0.31	−0.11	−0.24
	-	0.001	-	0.003	0.001	0.003	-	0.04
Total PLs	0.09	−0.19	−0.02	−0.19	0.33	0.16	0.10	0.19
	-	-	-	-	0.004	-	-	-
Total GPLs	0.07	−0.09	−0.15	−0.18	0.19	0.16	0.13	0.19
	-	-	-	-	-	-	-	-
PC	−0.11	0.01	−0.03	−0.01	0.10	−0.08	−0.09	−0.10
	-	-	-	-	-	-	-	-
LysoPC	−0.16	0.05	0.01	0.18	−0.28	−0.18	−0.17	−0.36
	-	-	-	-	0.02	-	-	0.001
PE	−0.15	−0.26	−0.22	−0.31	0.11	0.04	−0.05	−0.03
	-	0.02	0.04	0.006	-	-	-	-
PI	0.04	−0.04	−0.19	−0.25	0.10	0.12	0.12	0.21
	-	-	-	0.03	-	-	-	-
Rest GPLs	0.23	−0.07	−0.03	−0.11	0.11	0.25	0.25	0.32
	0.03	-	-	-	-	0.02	0.02	0.004
Ether GLs	−0.02	−0.03	−0.01	−0.10	0.05	−0.03	−0.02	0.11
	-	-	-	-	-	-	-	-
Plasmalogens	−0.21	0.26	−0.13	0.10	−0.23	−0.26	−0.15	−0.23
	0.04	0.01	-	-	0.04	0.01	-	-
Rest ether GLs	0.06	−0.13	0.05	−0.13	0.14	0.07	0.04	0.19
	-	-	-	-	-	-	-	-
Total SLs	0.07	−0.17	0.28	0.05	0.26	0.01	−0.06	−0.11
	-	-	0.008	-	0.02	-	-	-
SM	0.05	−0.39	−0.03	−0.23	0.26	0.14	−0.03	0.01
	-	0.001	-	-	0.02	-	-	-
Rest SLs	0.12	0.31	0.31	0.33	−0.08	−0.16	−0.02	−0.11
	-	0.003	0.003	0.004	-	-	-	-
Core Lipids	−0.05	0.17	0.01	0.17	−0.24	−0.11	−0.06	−0.14
	-	-	-	-	0.03	-	-	-
Surface Lipids	0.05	−0.17	−0.01	0.17	0.24	0.11	0.06	0.14
	-	-	-	-	0.03	-	-	-
CE/TC	0.10	−0.20	−0.04	−0.17	0.32	0.19	0.12	0.18
	-	0.04	-	-	0.005	-	-	-
CE/FC	0.08	−0.23	−0.04	−0.19	0.33	0.18	0.10	0.18
	-	0.03	-	-	0.004	-	-	-
CE/TG	0.08	−0.62	−0.07	−0.33	0.41	0.33	0.11	0.21
	-	0.001	-	0.004	0.001	0.001	-	-
TC/PLs	−0.09	0.05	0.001	0.12	−0.26	−0.09	−0.08	−0.14
	-	-	-	-	0.03	-	-	-
PC/SM	0.22	−0.02	0.16	0.28	−0.06	0.15	0.14	−0.01
	0.04	-	-	0.02	-	-	-	-

*: r Pearson Correlation, *p* value; -: not significant.

**Table 5 ijms-24-01995-t005:** Effect of gender on the HDL major lipid classes in the healthy population.

Major Lipid Classes	Total			30–39 Years		40–49 Years		50–59 Years		≥60 Years	
	Males	Females	*p*	Males	Females	Males	Females	Males	Females	Males	Females
n	44	46		14	13	11	12	8	7	11	14
Cholesterol, Total	41.71 ± 1.32	42.08 ± 1.81	NS	41.71 ± 0.72	41.14 ± 0.83	41.73 ± 1.28	42.01 ± 1.44	42.60 ± 1.75	43.06 ± 1.87	41.03 ± 1.39	42.52 ± 2.40
Free	7.85 ± 0.73	8.03 ± 1.24	NS	7.37 ± 0.33	6.76 ± 0.15 ***	7.66 ± 0.55	7.74 ± 0.48	8.73 ± 1.01	8.97 ± 0.72	8.00 ± 0.36	8.98 ± 1.31 *
Esterified	33.86 ± 1.01	34.05 ± 1.23	NS	34.34 ± 0.54	34.38 ± 0.74	34.07 ± 1.04	34.27 ± 1.18	33.87 ± 0.93	34.09 ± 1.46	33.03 ± 1.10	33.54 ± 1.47
Triglycerides (TG)	4.06 ± 0.74	4.10 ± 0.97	NS	3.75 ± 0.57	3.43 ± 0.37	3.75 ± 0.60	3.45 ± 0.42	4.19 ± 0.73	4.43 ± 0.89	4.69 ± 0.70	5.12 ± 0.76 *
Phospholipids (PLs), Total	54.23 ± 1.32	53.82 ± 2.19	NS	54.54 ± 0.69	55.43 ± 0.79 **	54.52 ± 1.55	54.54 ± 1.77	53.21 ± 1.38	52.51 ± 2.17	54.28 ± 1.43	52.36 ± 2.21 *
Core Lipids, Total	37.92 ± 0.99	38.15 ± 1.35	NS	38.08 ± 0.65	37.81 ± 0.73	37.81 ± 1.32	37.72 ± 1.50	38.05 ± 0.77	38.51 ± 1.81	37.72 ± 1.20	38.66 ± 1.34 *
Surface Lipids, Total	62.08 ± 0.99	61.85 ± 1.35	NS	61.92 ± 0.64	62.19 ± 0.73	62.19 ± 1.32	62.28 ± 1.50	61.95 ± 0.77	61.49 ± 1.81	62.28 ± 1.20	61.34 ± 1.34 *
Ratios											
CE/TC	0.81 ± 0.01	0.81 ± 0.02	NS	0.82 ± 0.00	0.84 ± 0.00 ***	0.82 ± 0.01	0.82 ± 0.01	0.80 ± 0.02	0.79 ± 0.01	0.81 ± 0.00	0.79 ± 0.02 *
CE/FC	4.35 ± 0.37	4.33 ± 0.62	NS	4.66 ± 0.18	5.08 ± 0.11 ***	4.47 ± 0.32	4.44 ± 0.25	3.91 ± 0.36	3.81 ± 0.28	4.13 ± 0.12	3.79 ± 0.46 *
CE/TG	8.63 ± 1.68	8.74 ± 1.96	NS	9.37 ± 1.52	10.13 ± 1.21	9.32 ± 1.54	10.08 ± 1.25	8.35 ± 1.66	7.95 ± 1.51	7.19 ± 1.11	6.70 ± 1.08
TC/PLs	0.77 ± 0.04	0.78 ± 0.07	NS	0.76 ± 0.02	0.74 ± 0.02 *	0.77 ± 0.04	0.77 ± 0.05	0.80 ± 0.05	0.82 ± 0.07	0.76 ± 0.04	0.82 ± 0.08 *

* *p* < 0.05; ** *p* < 0.01; *** *p* < 0.001: compared to males. NS: Not significant.

## Data Availability

Not applicable.

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
