# Peer review of "Effect of Clinical and Laboratory Parameters on HDL Particle Composition"

_ijms, 2023, doi:10.3390/ijms24031995_

Round 1
Reviewer 1 Report
Enclosed are some questions and suggestions:
- I would change ref.2 and ref.7 to a much more updated studies, given there are several papers demonstrating the importance of HDL structure and composition for their bioactivity.
- Lines 46-55: I would separate two diferrent ideas: the need of a pattern of HDL normal lipidome versus lipidome under pathological conditions, and the fact that plasma and serum can be the source of the HDLs used in the assays.
- Do the authors have any data using plasma in comparison to serum as a source of HDLs?
- Line 59: changes in lipidome are important for other HDLs activities, please add more references about this. (Kontush, A., Lhomme, M., & Chapman, M. J. (2013). Unraveling the complexities of the HDL lipidome1. Journal of lipid research, 54(11), 2950-2963).
- Any experiment related to cholesterol efflux, antioxidative capacity, anti-inflammatory or vasodilatory activity. known to be affected when HDL lipidome is altered? The same reference could be useful.
- Lines 61-62: Any previous clinical data using NMR for lipidome analyses? (Wang, D., Yu, B., Li, Q., Guo, Y., Koike, T., Koike, Y., ... & Zeng, R. (2022). HDL quality features revealed by proteome‒lipidome connectivity are associated with atherosclerotic disease. Journal of Molecular Cell Biology, 14(3), mjac004.)
- Lines 61-62: Which are the advantages of NMR for lipidome studies? (Wang, D., Yu, B., Li, Q., Guo, Y., Koike, T., Koike, Y., ... & Zeng, R. (2022). HDL quality features revealed by proteome‒lipidome connectivity are associated with atherosclerotic disease. Journal of Molecular Cell Biology, 14(3), mjac004.)
- Line 71-73: please clarify why apoAI and apoB are important data for HDL bioactivity.
- Line 106: please refer to the importance of the lenght of the fatty acids chains in some HDLs activities.
- Line 108: I would recommend some kind of scheme or graph to better understand tables 2, 5, 6 and 8 data.
- Tables 5-8 are not properly formatted.
- Please update some references, eg: 13-15, 17, 18, 20, 23, 24, 33, 36, 37, 39.
- Discussion includes a comparaison with a study dated 2007 (Berrougui, 2007) wich may not be appropiate to support the results.
- I miss more information about the influence of our diet on the lipidome, given previous data of HDL remodelling in line with this (Grao-Cruces, E., Santos-Mejias, A., Ortea, I., Marquez-Paradas, E., Martin, M. E., Barrientos-Trigo, S., ... & Montserrat-de la Paz, S. (2022). Proteomic analysis of postprandial high-density lipoproteins in healthy subjects. International Journal of Biological Macromolecules). (Grao-Cruces, E., Varela, L. M., Martin, M. E., Bermudez, B., & Montserrat-de la Paz, S. (2021). High-density lipoproteins and mediterranean diet: A systematic review. Nutrients, 13(3), 955.) (Zhu, C., Sawrey-Kubicek, L., Beals, E., Hughes, R. L., Rhodes, C. H., Sacchi, R., & Zivkovic, A. M. (2019). The HDL lipidome is widely remodeled by fast food versus Mediterranean diet in 4 days. Metabolomics, 15(8), 1-11.)
Author Response
Reviewer 1
We thank the reviewer for the kind consideration of our work and his very constructive comments. Our responses are as follow:
- I would change ref.2 and ref.7 to a much more updated studies, given there are several papers demonstrating the importance of HDL structure and composition for their bioactivity.
According to the reviewer’s suggestion, ref.2 and ref. 7 were updated (the new ref. 2 and ref. 7 in the revised manuscript in References Section are colored in blue).
- Lines 46-55: I would separate two different ideas: the need of a pattern of HDL normal lipidome versus lipidome under pathological conditions, and the fact that plasma and serum can be the source of the HDLs used in the assays.
We thank the reviewer for the comment. In the revised manuscript, we modified the text in the Lines 48-51 on Page 2. The parameter, biological fluid used in the assays (i.e., serum or plasma) that could affect HDL lipidome was included in the extrinsic factors.
- Do the authors have any data using plasma in comparison to serum as a source of HDLs?
No data are available concerning the comparison of the blood-derived matrices as a source of HDLs.
- Line 59: changes in lipidome are important for other HDLs activities, please add more references about this. (Kontush, A., Lhomme, M., & Chapman, M. J. (2013). Unraveling the complexities of the HDL lipidome1. Journal of lipid research, 54(11), 2950-2963).
According to the reviewer’s suggestion, more references were added.
- Any experiment related to cholesterol efflux, antioxidative capacity, anti-inflammatory or vasodilatory activity. known to be affected when HDL lipidome is altered? The same reference could be useful.
We thank the reviewer for this interesting point. It is well known that the antiatherogenic functional biological activities of HDLs directly reflect their lipid compositional characteristics (PMID: 23543772). However, in our study, it was not feasible to perform laboratory assays for the measurement of cholesterol efflux and antioxidative capacity as well as of anti-inflammatory or vasodilatory activity.
- Lines 61-62: Any previous clinical data using NMR for lipidome analyses? (Wang, D., Yu, B., Li, Q., Guo, Y., Koike, T., Koike, Y., ... & Zeng, R. (2022). HDL quality features revealed by proteome‒lipidome connectivity are associated with atherosclerotic disease. Journal of Molecular Cell Biology, 14(3), mjac004.)
According to the reviewer’s suggestion, more references on the NMR-based lipidomic analysis of clinical data were added.
- Lines 61-62: Which are the advantages of NMR for lipidome studies? (Wang, D., Yu, B., Li, Q., Guo, Y., Koike, T., Koike, Y., ... & Zeng, R. (2022). HDL quality features revealed by proteome‒lipidome connectivity are associated with atherosclerotic disease. Journal of Molecular Cell Biology, 14(3), mjac004.)
High-field NMR spectroscopy is a powerful and reliable tool to assess the molecular lipid composition of biological samples. It is a non-destructive approach with high analytical reproducibility that does not require extensive steps for sample preparation and offers easy identification of molecular lipid moieties and direct quantitative information in only one experiment. It is important to mention that the sample can be reused for additional NMR experiments or even recovered for other experimental modules.
Even though it was not used in this study, 2D NMR techniques (e.g. homo/hetero-nuclear COSY, HSQC) provide advanced structural elucidation of the lipid signals, can partially mitigate the relatively low sensitivity of NMR. Proton NMR (1H-NMR) provides quantitative screening of major lipid classes and has increasingly been used in lipidomic assessments as a rapid, simple and non-invasive detection method. NMR lipidomics also benefits of the fully quantitative nature of the technique with a high degree of consistency and reliability.
In the revised manuscript, we added the advantages of NMR for lipidomic studies (Page 2, Lines: 67-71).
- Line 71-73: please clarify why apoAI and apoB are important data for HDL bioactivity.
In clinical practice, the measurement of the conventional serum lipid parameters (total cholesterol, HDL-C, LDL-C and TGs) together with the determination of apoAI and apoB levels are widely used in screening programs of large populations to improve information about the compositional characteristics and metabolism of lipoprotein particles. As such, apoAI and apoB measurements were included in our study.
Especially for apoAI, data supports its determinant role on the functional properties of HDLs. As discussed in the Discussion Section, the conformation of apoAI, appears to be very sensitive to the nature of HDL core neutral lipids. More specifically, the α-helix stability of apoAI is enhanced by cholesterol esters but reduced by triglycerides, whereby apoAI dissociates from HDL and is cleared from the plasma. Also, the conformation of the central and C-terminal domains of apoAI is critical for HDLs to act as an acceptor of LDL-derived oxidized lipids, thus affecting their antioxidant activity. Since apoAI is the principle catalytic activator of lecithin-cholesterol acyl-transferase (LCAT), the disproportionation into cholesterol esters and triglycerides in HDL core possibly negatively influences the LCAT-mediated conversion of free cholesterol to cholesterol esters.
- Line 106: please refer to the importance of the length of the fatty acids chains in some HDLs activities.
Although the effect of the fatty acid saturation/unsaturation on HDLs’ functional properties has been investigated (Nicholls SJ et al. 2006, Davidson WS et el. 1995), however, there are no data in the literature concerning the influence of fatty acyl chain length on HDL biological activities.
Nicholls SJ, Lundman P, Harmer JA, Cutri B, Griffiths KA, Rye KA, Barter PJ, Celermajer DS. Consumption of saturated fat impairs the anti-inflammatory properties of high-density lipoproteins and endothelial function. J. Am. Coll. Cardiol. 2006, 48, 715–720.
Davidson WS, Gillotte KL, Lund-Katz S, Johnson WJ, Rothblat GH, Phillips MC. The effect of high density lipoprotein phospholipid acyl chain composition on the efflux of cellular free cholesterol. J. Biol. Chem. 1995, 270, 5882–5890.
- Line 108: I would recommend some kind of scheme or graph to better understand tables 2, 5, 6 and 8 data.
According to the reviewer’s suggestion, data on Tables 5 and 6 are illustrated in Figure 1 and those on Table 8 in Figures 2 and 3, in the revised manuscript.
- Tables 5-8 are not properly formatted.
We thank the reviewer. Changes were made in the revised manuscript.
- Please update some references, eg: 13-15, 17, 18, 20, 23, 24, 33, 36, 37, 39.
According to the reviewer’s suggestion, the references were updated.
- Discussion includes a comparison with a study dated 2007 (Berrougui, 2007) which may not be appropriate to support the results.
We thank the reviewer. We modified the text accordingly.
- I miss more information about the influence of our diet on the lipidome, given previous data of HDL remodelling in line with this (Grao-Cruces, E., Santos-Mejias, A., Ortea, I., Marquez-Paradas, E., Martin, M. E., Barrientos-Trigo, S., ... & Montserrat-de la Paz, S. (2022). Proteomic analysis of postprandial high-density lipoproteins in healthy subjects. International Journal of Biological Macromolecules). (Grao-Cruces, E., Varela, L. M., Martin, M. E., Bermudez, B., & Montserrat-de la Paz, S. (2021). High-density lipoproteins and mediterranean diet: A systematic review. Nutrients, 13(3), 955.) (Zhu, C., Sawrey-Kubicek, L., Beals, E., Hughes, R. L., Rhodes, C. H., Sacchi, R., & Zivkovic, A. M. (2019). The HDL lipidome is widely remodeled by fast food versus Mediterranean diet in 4 days. Metabolomics, 15(8), 1-11.)
Very interesting point. This can be considered a limitation of our study since no data on the diet consumed by our participants are available.

Reviewer 2 Report
The Authors focused on a study of the Effect of clinical and laboratory parameters on HDL particle composition. This is an interesting and comprehensive study. The article is well structured, but some parts are missing or not clear enough.
In my opinion:
- The abstract presents an accurate description of this study.
- The Authors was conducted adequate literature review.
- The references support the rationale for reporting the study.
- The patients are described adequately.
- The management of the study is effectively described.
- Valid and reliable outcome measures are utilized.
- The conclusions are appropriate.
- Tables are not clear enough, please modify and edit.
- The "Discussion" section should be rewritten showing correlations in a more organized way.
- The entire article should be edited once again, as in this form it is very chaotic and unreadable.
Sometimes there is a period at the end of the table description, and sometimes there is not, please decide on one of the versions and correct throughout the article.
Author Response
Reviewer 2
The Authors focused on a study of the Effect of clinical and laboratory parameters on HDL particle composition. This is an interesting and comprehensive study. The article is well structured, but some parts are missing or not clear enough.
In my opinion:
- The abstract presents an accurate description of this study.
- The Authors was conducted adequate literature review.
- The references support the rationale for reporting the study.
- The patients are described adequately.
- The management of the study is effectively described.
- Valid and reliable outcome measures are utilized.
- The conclusions are appropriate.
- Tables are not clear enough, please modify and edit.
- The "Discussion" section should be rewritten showing correlations in a more organized way.
- The entire article should be edited once again, as in this form it is very chaotic and unreadable.
Sometimes there is a period at the end of the table description, and sometimes there is not, please decide on one of the versions and correct throughout the article.
We thank the reviewer for the kind consideration of our work and his/her very constructive comments. Tables were modified accordingly, and data were also represented in figures. The discussion section was rewritten in a more organized way.

Round 2
Reviewer 2 Report
Accept in present form.